Enhancing brain tumor MRI classification with an ensemble of deep learning models and transformer integration

Benzorgat Nawal nawel.benzorgat@gmail.com
Xia Kewen kwxia@hebut.edu.cn
Benzorgat Mustapha Noure Eddine
School of Electronics and Information Engineering, Hebei University of Technology , Tianjin , China
Cirillo Stefano
Electronic publication date: 2024 Nov 27
Publication date: 2024
Volume: 10
Electronic Location ID: e2425
Received 2024 Jun 10; Accepted 2024 Sep 26
Copyright: ©2024 Benzorgat et al.
Copyright year: 2024
Copyright holder: Benzorgat et al.
License: This is an open access article distributed under the terms of the Creative Commons Attribution License, which permits unrestricted use, distribution, reproduction and adaptation in any medium and for any purpose provided that it is properly attributed. For attribution, the original author(s), title, publication source (PeerJ Computer Science) and either DOI or URL of the article must be cited.
License URL: https://creativecommons.org/licenses/by/4.0/

Keywords: Brain tumor detection, Deep learning, Transfer learning, Transformer encoder, Shifted window-based self-attention, Multilayer perceptron (MLP), MRI data analysis, Ensemble learning, Pretrained models, Medical imaging

Funding: National Natural Science Foundation of China No. 42075129 Hebei Province Natural Science Foundation No. E2021202179 Key Research and Development Project from Hebei Province No. 21351803D Hebei special project for key technology and product R&D No. SJMYF2022Y06 This work was supported by the National Natural Science Foundation of China (No. 42075129), the Hebei Province Natural Science Foundation (No. E2021202179), the Key Research and Development Project from Hebei Province (No. 21351803D) and the Hebei special project for key technology and product R&D (No. SJMYF2022Y06). The funders had no role in study design, data collection and analysis, decision to publish, or preparation of the manuscript.

==============================
Brain tumors are widely recognized as the primary cause of cancer-related mortality globally, necessitating precise detection to enhance patient survival rates. The early identification of brain tumor is presented with significant challenges in the healthcare domain, necessitating the implementation of precise and efficient diagnostic methodologies. The manual identification and analysis of extensive MRI data are presented as a challenging and laborious task, compounded by the importance of early tumor detection in reducing mortality rates. Prompt initiation of treatment hinges upon identifying the specific tumor type in patients, emphasizing the urgency for a dependable deep learning methodology for precise diagnosis. In this research, a hybrid model is presented which integrates the strengths of both transfer learning and the transformer encoder mechanism. After the performance evaluation of the efficacy of six pre-existing deep learning model, both individually and in combination, it was determined that an ensemble of three pretrained models achieved the highest accuracy. This ensemble, comprising DenseNet201, GoogleNet (InceptionV3), and InceptionResNetV2, is selected as the feature extraction framework for the transformer encoder network. The transformer encoder module integrates a Shifted Window-based Self-Attention mechanism, sequential Self-Attention, with a multilayer perceptron layer (MLP). These experiments were conducted on three publicly available research datasets for evaluation purposes. The Cheng dataset, BT-large-2c, and BT-large-4c dataset, each designed for various classification tasks with differences in sample number, planes, and contrast. The model gives consistent results on all three datasets and reaches an accuracy of 99.34%, 99.16%, and 98.62%, respectively, which are improved compared to other techniques.

Introduction

The brain is a complex organ that functions through the coordination of billions of cells. Brain tumors arise from the uncontrolled proliferation of these cells, forming abnormal clusters in or around the brain (Kibriya et al., 2022). Tumors can be malignant, with cancerous properties and the ability to spread, or benign, which do not spread. The incidence of malignant brain tumors increases with age (Ostrom et al., 2018). Malignant tumors are characterized by rapid and uncontrolled growth (referred to as high grade) and have indistinct borders. These tumors can either start in the brain and extend to other parts of the body, known as primary malignant tumors, or they can originate in other areas of the body and spread to the brain, termed secondary malignant tumors (Sharif et al., 2022; Mohsen, et al., 2018).

There are three types of primary brain tumors: meningiomas, gliomas, and pituitary tumors; Fig. 1 illustrates this three main types. These tumors arise from brain tissues or their immediate surroundings. Gliomas are cancerous tumors that develop in the glial cells, which support nerve cells in the central nervous system (brain or spinal cord). Meningiomas grow on the meninges, the three-layer membranes covering the brain and spinal cord. Pituitary tumors form around the pituitary gland inside the skull, which regulates hormone levels in the body (Nallamolu et al., 2022). Among these, gliomas are the most common slow-growing malignant tumors in the brain. Although they rarely spread to the spinal cord or other organs, they can grow into different parts of the brain, making them potentially life-threatening (Louis et al., 2016). Early detection is crucial for effective treatment. MRI and CT scans are essential diagnostic tools, with MRI being more beneficial due to its ability to detail the shape, location, and size of tumors.

In neuroscience, timely brain tumor detection is critical for saving lives. Despite various existing methodologies for detecting anomalies in MRI scans, there is room for improvement in efficiency and rapid classification (Mandle, Sahu & Gupta, 2022). Traditional methods struggle with the increasing volume of medical data, necessitating computerized support technologies. Artificial Intelligence (AI), especially deep learning and machine learning, has shown significant success in visual tasks and is increasingly used for early illness detection, driving researchers to refine current approaches (Brutonet al., 2020).

Deep learning is extensively used to analyze biomedical data in healthcare (Talukder et al., 2022), with recent advancements achieving successful brain tumor classification. Deep learning can automatically extract meaningful features and handle large quantities of data. Convolutional neural networks (CNNs) are particularly effective in image processing tasks, including enhancement, generation, detection, segmentation, classification, and recognition. CNNs outperform traditional methods by automatically extracting features from input images. Implementing deep learning techniques for tumor detection in MRI can reduce radiologists’ workload, but the diversity of deep learning approaches creates uncertainty about the best model to use. Additionally, the focus on single CNN models for brain tumor identification raises questions about the potential of ensemble learning methods to improve classification accuracy.

CNN models assess the spatial relationships between neighboring pixels within a defined receptive region (Al-masni et al., 2018), a challenge overcome by integrating attention mechanisms. Attention helps identify and focus on the most informative data segments. While deep learning effectively captures local features, it struggles with extracting global features for long-range dependencies. The self-attention mechanism in Vision Transformers (ViTs) addresses this, effectively modeling long-range dependencies for accurate brain tumor classification.

This study proposes a new method using CNN models to detect binary and multiclass brain tumors in MRI images, combining a Transformer Encoder with a Self-Attention Network and a multilayer perceptron. Unlike previous studies that enhanced MRI images for better results, our framework processes original images directly, eliminating the need for prior enhancements. The focus is on advanced transformer encoder methodologies for feature extraction from unprocessed images, rather than traditional CNN models. The Transformer model, with its self-attention structure, has improved concurrent processing and translation accuracy (Shamshad et al., 2023). In contrast, CNN models, with fixed filter sizes, struggle to capture relationships at lower resolutions. Vision Transformers (ViTs) treat objects as sequences, allowing algorithms to understand image hierarchies and infer class labels independently (Al-masni et al., 2018).

Our research does not incorporate image enhancement techniques except for data augmentation, unlike other studies. In our model, input images are treated as sequences of patches, each condensed into a single feature vector, allowing for detailed feature examination on a patch-by-patch basis, enhancing performance. We evaluated six pre-trained models EfficientNetB7, VGG16, DenseNet201, InceptionResNetV2, GoogleNet, and Xception both individually and in combination. These models serve as feature extractors for the transformer encoder. We introduce an automated technology capable of identifying and differentiating various brain tumor types, assisting in medical diagnosis when trained radiologists are unavailable. Using three publicly available datasets, we evaluate our network against baseline models and recent research, demonstrating that our method outperforms other advanced methods. The contributions of the paper are enumerated as follows:

(1) An improved hybrid CNNs model that emphasizes brain tumors detection diseases in MRI images in the early stages based on the ensemble learning and transformer encoder.

(2) A thorough and robust examination of classifying brain tumors is conducted using three public MRI datasets: the Cheng dataset (Cheng, 2017), BT-large-2c (Hamada, 2021) and BT-large-4c dataset (Satarj, 2020).

(3) Ensemble deep learning-based feature extraction framework which proves highly effective in distinguishing different types of brain tumor diseases. This analysis is carried out in both binary and multiclass classification scenarios.

(4) A Bayesian Optimization is applied to automate the tuning process of finding the optimal set of hyperparameters for transformer encoder and classification layers to fine tune the model, aiming to improve its performance on all the datasets.

(5) Examining the categorization effectiveness of various pretrained deep learning models, both individually and when combined into ensembles.

The following sections structure this article: Section 2, denoted as “Related work”, provides a detailed overview of relevant studies. Section 3, titled “An improved method”, outlines the proposed method, and dataset used in our research. Section 4 designated as “Experiment and result analysis” covers the experimental objective, parameters setting, performance evaluation and the results discussions about the proposed method. Section 5, entitled “Conclusion”, presents the concluding potential future research and remarks.

Related Work

Deep learning excels in classification and detection, impacting medical image analysis. It has shown consistent success in various challenges, especially in disease identification. The related research on brain tumor diagram representing classification shows in Fig. 2.

Figure 1 The illustration of the three main types of tumors.

Figure 2 The tree diagram of related works for brain tumor classification.

(1) Classification using CNNs

Badza & Barjaktarovic (2020) introduced a novel CNN model for brain tumor classification based on an enhanced version of the AlexNet structure. The model was evaluated using the Cheng dataset, and its generalization ability was systematically assessed through various methods, including 10-fold cross-validation. The record-wise cross-validation for augmented data yielded superior results, achieving an impressive accuracy rate of 96.56%.

Similarly, Ayadi et al. (2021) presented an innovative CNN model with multiple layers designed for categorizing MRI brain tumors. Their approach demonstrated superior recognition and classification accuracy compared to previous studies using the same data, achieving accuracy rates of 94.74% for the Cheng dataset, 93.71% for the Radiopaedia dataset, and 97.22% for the Rembrandt dataset. In another study, Salama & Shokry (2022) proposed two deep models for the detection and classification of brain tumors. The first model, a convolutional generative model, addressed a small, class-imbalanced dataset, while the second focused on binary classification. Despite a slower training time compared to CNN and transfer learning models, the approach achieved an accuracy of 96.88% on the Chakrabarty N dataset.

Kalaiselvi et al. (2022) proposed a novel activation function, E-Tanh, which extends the Tanh function to improve ANN performance. E-Tanh outperformed existing activation functions like ReLU and Sigmoid, particularly in shallow models on the MNIST dataset. While it showed competitive results in CNN models, it was less effective in wide residual networks on CIFAR datasets.

In an effort to enhance CNN models, Amou et al. (2022) introduced an enhanced hyperparameter optimization strategy for CNNs based on Bayesian optimization. This method was assessed by classifying brain MRI scans into three cancer categories. They investigated five widely recognized deep pre-trained models to enhance CNN efficiency using transfer learning. Their CNN achieved a maximum accuracy rate of 98.70% after employing Bayesian optimization, surpassing the performance of previous studies and demonstrating the feasibility of automating hyperparameter optimization

(2) Classification of brain tumors using transfer learning (TL)

After AlexNet (Krizhevsky, Sutskever & Hinton, 2012) was introduced, CNNs became widely used in computer vision. Researchers have since developed more advanced architectures like VGG, GoogleNet, ResNet, DenseNet, and EfficientNet to improve deep learning in computer vision. Polat & Gungen (2021) examined transfer learning networks for classifying brain tumors from MRI images, achieving 99.02% accuracy with ResNet50 and Adadelta optimization on the Cheng dataset.

Ismael, Mohammed & Hefny (2020) used a ResNet to categorize brain tumors into three classes, achieving 99.0% accuracy on the Cheng dataset. Their study highlighted the potential of transfer learning in medical diagnoses.

In the research conducted by Rehman et al. (2020), three CNN architectures were employed in experiments to diagnose brain lesions using MRI slices from the Cheng dataset. They used transfer learning techniques like fine-tuning and freezing, along with data augmentation, to improve generalization and reduce overfitting. The fine-tuned VGG16 model achieved a prediction rate of 98.69%.

Swati et al. (2019) introduced a blockwise fine-tuning method using transfer learning with a pre-trained CNN model, achieving an average accuracy of 94.82% on the Cheng dataset. Their VGG19 technique outperformed conventional CNN-based methods. Sajjad et al. (2019) used a pre-trained CNN model fine-tuned with augmented data to classify brain tumors with various grades. The model achieved accuracy rates of 94.58% on the Brain Tumor dataset and 90.67% on the Radiopaedia dataset.

In a separate study, Sharif et al. (2020) developed a deep learning system for the segmentation and classification of brain tumors using a saliency-based deep learning technique. Their model, which fine-tuned the Inception-V3 model, achieved 92% accuracy in brain tumor classification with evaluations conducted on BraTS 2018, BraTS 2017, BraTS 2014, and BraTS 2013 datasets.

(3) Brain tumor classification using Vision Transformers (ViT)

Numerous researchers have focused on improving CNNs, achieving significant advancements. However, a CNN model that excels on specific datasets may underperform on others due to its focus on analyzing correlations among spatially adjacent pixels, limiting its ability to establish connections with distant pixels. To address this, recent research has incorporated attention mechanisms to enhance performance by identifying and focusing on the most informative data segments.

Hu et al. (2020) introduced SENet, which enhances classification performance by reducing superfluous information and accurately weighting image channels with minimal computational cost, achieving a 25% relative improvement and a top-5 error rate of 2.251% in ILSVRC 2017.

Zhang & Yang (2021) developed the Shuffle Attention mechanism, which combines channel and spatial attention, improving accuracy by 1.34% in ImageNet-1k classification and effectively capturing small lesions in the medical domain. It outperforms state-of-the-art methods on benchmarks like ImageNet-1k and MS COCO, achieving higher accuracy with lower complexity.

Tummala et al. (2022) utilized pre-trained and fine-tuned Vision Transformer (ViT) models for categorization tasks, achieving a total testing accuracy of 98.7% on the Cheng dataset by combining four ViT variants (B/16, L/16, B/32, L/32). In another approach, Xu & Prasanna (2022) introduced a deep anchor attention learning method integrating ViT, showing efficacy in classifying overall survival in brain tumor patients using MRI.

Ferdous et al. (2017) proposed the Linear-Complexity Data-Efficient Image Transformer (LCDEiT), which employs a teacher-student strategy and external attention mechanism to reduce computational complexity. The LCDEiT model achieved notable performance on Figshare and BraTS-21 datasets, with average classification accuracies and F1-scores of 98.11% and 97.86%, and 93.69% and 93.68%, respectively.

In a separate study, Aladhadh et al., (2022) suggested a ViT-based model for efficient skin cancer classification, yielding promising results. Their proposed method achieved a precision of 96.00%, recall of 96.50%, sensitivity of 97.00%, and an F1-Measure of 96.14% on the HAM10000 dataset, demonstrating its effectiveness compared to current state-of-the-art techniques for SC classification. Additionally, various researchers have used different ViT versions for segmenting brain tumors (Hatamizadeh et al., 2022).

Despite these advancements, individual CNN and ViT-based techniques have limitations in classifying brain tumors, highlighting the need for further improvement. Consequently, the implementation of a hybrid model combining the strengths of ViT and CNN for accurate brain tumor classification is proposed, as detailed in the forthcoming section.

The Improved Hybrid WinShift Transformer CNN Method

Improvement strategies

This section introduces our proposed methodology, Hybrid WinShift Transformer CNN (HWST), outlining the different TL algorithms employed in our approach.

Improvement strategy 1

First, we discuss the basic concept of our approach, followed by a detailed description of the suggested hybrid architecture, as seen in Fig. 3. This model is inspired by the combination of CNNs for their powerful feature extraction with transformer encoder methods. CNNs have long dominated computer vision modeling. As widely recognized, deep learning CNN architectures possess the capability to examine the spatial interconnection existing among adjacent pixels within the region of interest delineated by the size of the convolutional filter, while disregarding the directional associations influenced by the separation between these pixels (Shamshad et al., 2023). To overcome this limitation, attention-based transformers have recently emerged, demonstrating enhanced effectiveness and robustness in managing both spatial pixel correlation and distance relations, thus enhancing the accuracy of visual recognition tasks.

Figure 3 The proposed hyper deep learning Transformer Encoder architecture model for brain tumor classification.

Improvement strategy 2

Second, ViT demand extensive training datasets, a significant limitation in the field of medical image analysis (Liu, Lin & Cao, 2021). A hybrid approach is proposed that leverages the Swin Transformer architecture to mitigate this issue. The suggested hybrid block comprises the following modified layers: position embedding, patch partitioning, linear embedding, a Swin Transformer block, and patch merging. These layers are specifically adapted to process input features effectively.

Implementation of hybrid WinShift transformer CNN

Overall design of the method (with a block diagram and steps)

The suggested hybrid framework involves the following processing steps:

• Step one: Data pre-processing was conducted to prepare the data, including resizing, normalization, cropping, dividing the data into testing, training, and validation sets, and applying data augmentation techniques.

• Step two: Feature extraction was performed using advanced ensemble CNN architectures. To determine the optimal backbone network for the hybrid framework, an extensive experimental examination of six trained models was carried out.

• Step three: After determining the best hyperparameters using Bayesian Optimization, the combined features from the models were processed through the transformer encoder.

• Step four: The final results were produced by feeding the acquired features into the classification layer.

Algorithm implementation (with algorithm flowchart and pseudocode table)

The implementation algorithm of the HWST model is analyzed in this section. The pseudocode for the algorithm is provided in Table 1, and the corresponding flow chart is illustrated in Fig. 4.

Table 1 Improved algorithm pseudo code.

Algorithm of Deep Learning Model Construction	
SEQUENCE	
Input: datasets images	
Output: Type of brain tumor	
BEGIN	
1. Def pretrained models ():	
2. Disable the trainable layers in the pretrained models	
3. Zero-padded the uniformized the outputs of the pertained models	
4. Concatenate the outputs after the uniformization	
5. Position embedding Layer (input dimension (None, 12, 12, 5504))	
6. Patch Partition Layer (input dimension (None, 36, 22016))	
7. Linear Embedding layer	
8. Construct Swin Transformer blocks (input dimension (None, 36, 96))	
For i = 0 to 1	
if i % 2 == 0 THEN	
shift_size_temp = 0	
else	
shift_size_temp = window_size/2	
end if	
Add SwinTransformerBlock with specified parameters to the model	
Patch merging layer	
End for	
9. Flatten Layer	
10. Dense Layer (with 256 units, Set the activation function ReLU)	
11. Dropout layer (0.5)	
12. Dense Layer (with 96 units, Set the activation function ReLU)	
13. Dropout layer (0.1)	
14. Dense (with 3 units, Set the activation function to softmax)	
15. Compile model (with Adam optimizer, categorical_crossentropy as the loss function, and accuracy as a metric)	
END	
END SEQUENCE	

Figure 4 Flow chart of algorithm.

Data collection

Three different datasets were used to train, validate, and evaluate the proposed deep learning models. The Cheng dataset (Cheng, 2017) contains 3,064 T1-weighted contrast-enhanced images from 233 patients, with three types of brain tumors covered: glioma (1,426 slices), pituitary tumor (930 slices) and meningioma (708 slices). The BT-large-2c (Hamada, 2021) dataset consists of 3,000 brain MRI images and contains 1,500 normal images, and 1,500 are tumor images and BT-large-4c dataset (Satarj, 2020) consists of 3,264 brain MR images and contains 827 pituitary tumor images, 826 glioma images, and 822 mengioma images, while the remaining 395 are normal images. In the three datasets mentioned, the same split ratio is used 80% for training, 10% for validations and 10% used for unseen data. Figure 5 illustrates the visual representations of several MRI images obtained from various datasets.

Figure 5 Some MRI samples images from different datasets: (1) (a) Normal image from Brats-large-2C dataset, (1) (b) Tumor image from Brats-large-2C dataset, (2) (a) Meningioma image from Cheng dataset, (2) (b) Glioma image from Cheng dataset, (2) (c) Pituitary tumor image from Cheng dataset, (3) (a) Glioma image from Brats- large-4C dataset, (3) (b) Meningioma image from Brats-large-4C dataset, (3) (c) Normal image from Brats-large-4C dataset, (3) (d) Pituitary tumor image from Brats-large-4C dataset.

Data pre-processing

In the data pre-processing phase, we prepared the dataset by cropping images to remove unnecessary areas, ensuring that the model focuses only on relevant features (as shown in Fig. 6). The images were resized to 224 × 224 pixels to reduce computational load and enhance efficiency, with an example of a cropped image illustrated in Fig. 5. We scaled the images by dividing by 255 and partitioned the dataset into training, validation, and unseen sets, with proportions of 80%, 10%, and 10%, respectively. To minimize loss, reduce variance, and improve generalization, shuffling was applied. Additionally, data augmentation techniques such as shifts, shearing, zooming, and flipping were used to enhance the training data. The final dataset distribution is detailed in Table 2. This study focused on the immediate application approach, applying models directly to the entire input MRI images.

Figure 6 An MRI image before and after cropping.

Multi-model in ensembles

Deep learning networks, known for their non-linear nature, offer significant adaptability, especially when dealing with limited training datasets (Talukder et al., 2023). Their sensitivity to specific training data arises from fine-tuning with random algorithms, leading to variability in weight sets with each training iteration and introducing variance in neural network predictions.

To address this variance, recent efforts have employed ensemble learning techniques, which involve training multiple models instead of a single one (Zhang & Yang, 2021). Ensemble learning integrates predictions from multiple diverse models to derive the final outcome, leveraging the strengths of different models to improve overall performance. This approach captures a broader range of intricate and salient image features that might be missed by individual models, extracting additional valuable information since different models may specialize in identifying various aspects of the data. Consequently, this enhances the robustness of classification results.

Notably, prevailing deep learning methodologies for brain tumor identification typically rely on a single convolutional network, with limited exploration of ensemble learning in this domain. This study addresses this gap by incorporating ensemble models in the design of the backbone network. The proposed methodology strategically combines various pre-trained models to enhance feature extraction and performance, leveraging their unique strengths and complementary capabilities. Two distinct ensemble scenarios were tested to explore and maximize the benefits of ensemble learning.

Table 2 Details of the datasets used in this study.

No.	Dataset name	Classes	Number of each class	Total no. of images	
1	Cheng Dataset	Glioma	1,426	3,064	
		Mengioma	708		
Pituatry	930	
2	BT-large-4c	Glioma	826	3,264	
		Mengioma	822		
		Pituatry	827		
No tumor	395	
3	BT-large-2c	Tumor	1,500	3,000	
Non-tumor	1,500	

(1) Ensemble A

This ensemble combines the structures of DenseNet201, GoogleNet (InceptionV3), and InceptionResNetV2. The initial classification layer of each model is removed, and deep features are extracted from the last convolutional layer of each model. The output dimensions of DenseNet architecture are (None, 7, 7, 1920), while GoogleNet and InceptionResNetV2 produce outputs with dimensions of (None, 5, 5, 2048) and (None, 5, 5, 1536) respectively. Due to the dissimilar output dimensions from GoogleNet and InceptionResNetV2 architectures, it is necessary to standardize all output features by using zero-padding before concatenating them.

(2) Ensemble B

This ensemble combines the architectures of DenseNet201, GoogleNet (InceptionV3), and VGG16 convolutional networks. As with Ensemble A, the classification layer of each model was removed, and the output features were combined from their final convolutional blocks or layers. The output dimensions for DenseNet, GoogleNet, and VGG16 networks are configured as (None, 7, 7, 1920), (None, 5, 5, 2048), and (None, 7, 7, 512), respectively. Due to the different output dimensions, particularly from GoogleNet, zero-padding was applied to standardize the output features before concatenation.

Transformer encoder (fine-tuned ViT)

A transformer is an advanced model in machine learning that uses the self-attention mechanism to attribute different levels of significance to various elements within the input data. It is designed using an encoder–decoder framework. This study focuses on adapting and refining the Vision Transformer using an encoder approach for the early detection of brain tumors in MRI images. The transformer encoder proposed in this research comprises a Patch Partition layer, a Linear Embedding layer, a Swin Transformer Block, and a Patch Merging layer.

The Transformer encoder generates two outputs from a given input image: the comprehensive input token embedding vector, which includes both image and class tokens, and the attention weights from all layers and heads. The embedding tensor’s dimension is determined by the number of tokens. Since we are interested in using the embedding vector of the class token for use as the image feature vector in classification, we extracted it at index 0. This vector is passed through the classifier, which produces an output whose length corresponds to the number of classes. The following sections elaborate on the components of the fine-tuned model.

Swin transformer block

The Swin Transformer is created by replacing the standard multi-head self-attention (MSA) module within a Transformer block with a module that utilizes shifted windows, as explained in ‘Self-Attention Using Shifted Windows’, while keeping other layers unchanged. A Swin Transformer block consists of a shifted window-based MSA module followed by a 2-layer MLP with GELU non-linearity in between. Layer normalization (LN) is applied before each MSA module and each MLP, and a residual connection is applied after each module.

Self-attention using shifted windows

The traditional Transformer architecture and its adaptation for image classification (Dosovitskiy et al., 2020) implement global self-attention, where each token’s relationship with all other tokens is computed. This global computation results in quadratic complexity in relative to the number of tokens, making it inefficient for vision tasks that require a vast number of tokens for dense predictions or high-resolution image representation.

Non-overlapping window-based self-attention

To enhance modelling efficiency, this approach calculates self-attention within specific local windows. These windows are organized to evenly divide the image without overlapping. Assuming each window consists of M × M patches, the computational complexity of both a global MSA module and a window-based one on an image with h × w patches is:

(1) ΩMSA=4hwC2+2hw2C

(2) ΩW−MSA=4hwC2+2M2hwC.

The first equation shows that the complexity is quadratic in terms of the number of patches hw, making global self-attention computation impractical for large values of hw. In contrast, the latter equation demonstrates linear complexity when M is fixed (typically set to 7 by default). This design choice ensures that window-based self-attention remains scalable, providing a more feasible approach for handling large hw values.

MLP layer

The multilayer perceptron (MLP) is a neural network model known for its feed-forward design, which includes dense and dropout layers. In this study, the MLP features two non-linear layers that use Gaussian Error Linear Units (GELU). The MLP blocks are arranged in a consistent manner, with layers stacked in similar patterns. For example, let X ∈ ℝn×d denote token attributes, where n stands for the sequence length and d represents the dimension. Each block is mathematically described as:

(3) Z=σXU,Z ~=sZ,Y=Z ~V

(4) z0=xclass;xp1E;xp2E;⋯;xpNE+EPOS,E∈Rp2∗C∗D,Epos∈RN+1∗D,

(5) zlI=MSALNzl−1+zl−1,l=1…..L

(6) zl=MLPLNzIl+zlI,l=1…..L,

(7) y=LNzl0.

Classification layer

This study focuses on identifying brain tumors across multiple classes. Input samples are processed through the encoding network, which produces outputs that are converted into the probability range 0,1 using the Softmax layer. In this context, the bias term and the weight matrix are represented bc and Wc, respectively. The categorical cross-entropy function is employed to compute the loss between the ground truth and the predicted labels.

Experiment and Result Analysis

Experimental environment and datasets

All experiments were conducted using Kaggle’s computational resources. Kaggle provided 73.1 GB of disk space to efficiently store and manage the dataset, model checkpoints, and experimental results. The Kaggle environment offers 13 GB of RAM, crucial for data loading and manipulation, especially with large datasets and complex models. We had access to a GPU with 15.9 GB of memory, essential for efficient and fast deep learning model training, including the proposed model. Kaggle allocated 19.5 GB for output storage, allowing for the storage and analysis of model outputs, visualizations, and experimental data.

Evaluation indicators

Various performance metrics were used to assess the effectiveness of the proposed hybrid deep learning framework, including accuracy (ACC), precision (PRE), sensitivity (SEN), F1-score, and the area under the receiver operating characteristic (ROC) curve, as described by the following equations. (8) AccuracyACC=TP+TNTP+FN+TN+FP.

Where TP is True Positives, TN is True Negatives, FP is False Positives, and FN is False Negatives.

Precision (PRE), also known as the percentage of correctly predicted positive values, is defined as: (9) PrecisionPRE=TPTP+FP.

This metric measures the proportion of true positive predictions among all positive predictions made.

Sensitivity (SEN) also known as true positive rate, this metric indicates the proportion of actual positives correctly identified, and is mathematically represented as: (10) SensitivitySEN=TPTP+FN.

The F1-Score is the harmonic mean of precision and sensitivity, providing a balance between the two metrics, represented as: (11) F1-Score=2×TP2×TP+FP+FN.

The classifier’s performance is evaluated through the area under the curve (AUC), where a probability curve is generated by plotting the FP rate against different threshold settings, forming the receiver operating characteristic (ROC) curve. A higher AUC indicates better identification performance, with an AUC of 1 indicating perfect classification and an AUC of 0.5 suggesting random classification.

Result analysis

Hyperparameters

The HWST model is trained in an end-to-end manner using specific hyperparameters to enhance its performance. The training parameters include a learning rate of 0.0001, which is reduced by a factor of 0.2, an epsilon value of 0.00001, and a patience of 7. Additionally, an early stopping strategy with a patience of 10 is implemented. This strategy can execute operations at multiple phases of learning, including different batch and epoch intervals.

The model is compiled with the Adam optimizer, a clip value of 0.2, and an epoch count of 100. During the selection of pre-trained models, an epoch of 100 is maintained while other parameters remain constant. The encoder transformer block is configured with a patch size of 2, a window size of 2, and global average pooling (GAP) for the shift size. Bayesian optimization, specifically using the Optuna framework, is employed to optimize additional parameters. These include dropout rates after each MLP layer, dropout rates after Swin-Attention, dropout at the end of each Swin-Attention block, drop-path regularization within skip connections, the number of attention heads, embedded dimensions, and nodes in the MLP layers.

This systematic optimization ensures an empirically refined configuration, enhancing the performance and adaptability of the encoder transformer block within the specified framework. The values of these parameters are detailed in Table 3. The output from the transformer encoder block is fed into a dense neural network consisting of two layers: the first with 256 nodes and a dropout rate of 0.5, and the second with 96 nodes and a dropout rate of 0.1. These specific values were determined through the hyperparameter optimization process.

This section provides an overview and explanation of the results obtained from the proposed models. Initially, we examined the sensitivity of the models’ parameters and reported the research findings accordingly. The classification performance of the models, both individually and in combination, is then presented and discussed. Table 4 summarizes the comparison of all models employed in this study, focusing on their final output shape and the number of trainable and non-trainable parameters.

Table 3 Training hyper-parameter values of proposed transformer.

Hyper-parameter	Value	
Dropout after each MLP layer	0.2995	
Dropout after Swin-Attention	0.7605	
Dropout at the end of each Swin-Attention block	0.5855	
Drop-path within skip-connections	0.4789	
Number of attention heads	4	
Number of embedded dimensions	96	
Number of MLP nodes	24	

Table 4 Parameters of the proposed models, including the final output shape and the number of trainable and non-trainable parameters.

Models	Last convolution layer output	Trainable parameter	Non-trainable parameter	Total parameter	
DenseNet201	None, 7, 7, 1920	5,724,083	18,322,160	24,046,243	
VGG16	None, 7, 7, 512	3,561,395	14,714,864	18,276,259	
GoogleNet	None, 5, 5, 2048	5,920,691	21,802,960	27,723,651	
InceptResNetV2	None, 5, 5,1536	5,134,259	54,336,912	59,471,171	
Xception	None, 7, 7, 2048	5,920,691	20,861,656	26,782,347	
Ensemble A	None, 7,7,6528	209,027	103,281,144	103,490,171	
Ensemble B	None, 7, 7, 6528	209,027	104,222,448	104,431,475	
Proposed hybrid model with Ensemble A backbone	–	12,801,971	103,281,320	116,083,291	
Proposed hybrid model with Ensemble B backbone	–	12,801,971	104,222,624	117,024,595	

Classification results

Before implementing the HWST model, a thorough analysis of parameter sensitivity was conducted to determine the optimal number of ensemble deep learning models. Various ensemble configurations, involving 1, 2, 3, and 4 deep learning models, were tested. The combinations of ensemble’s deep learning models were randomly selected from six options: DenseNet201, VGG16, GoogleNet, InceptionResNetV2, Xception, and EfficientNetB7.

The outcomes of the multiclass classification scenario, as shown in Table 5, present the classification evaluations based on overall identification accuracy (ACC), precision (PRE), and F1 score. The most favorable classification performance of the proposed model was observed when with three ensemble feature extractor models.

Table 5 The best result of the combination deeps learning model’s options.

Number of pre-trained ensemble models	ACC	PRE	F1-score	
One Model	0. 9738	0.8865	0.9223	
Two Models	0.9804	0.9329	0.9478	
Three Models	0.9870	0.9860	0.9947	
Four Models	0. 9853	0.9231	0.9514	

Subsequently, the transformer encoder was applied to the best-performing ensemble, incorporating hyperparameter optimization using the Bayesian optimizer. Table 3 details the specific parameters obtained from this optimization along with the corresponding search results.

This section presents the outcomes of the classification approaches implemented in this article. The model’s classification performance was evaluated across three scenarios: individual pre-trained models, ensemble models know as Ensemble A and Ensemble B, and the proposed hybrid model based on the ensemble transformer encoder mechanism. All experiments in this study utilized the Cheng dataset for the multiclass classification task using MRI images. This dataset was chosen for its reliability and its ability to demonstrate the models’ capacity to handle multiple classes concurrently.

(1) Individual pre-trained models classification results

Six state-of-the-art deep learning models were selected for the backbone ensemble network: DenseNet201, Xception, VGG16, GoogleNet, InceptionResNetV2, and EfficientNetB7. The individual classification performance metrics for each model were recorded and presented in Table 6. The models were trained under the same settings and system setup for consistency. Among these models, DenseNet201 demonstrated superior performance in terms of accuracy, sensitivity, precision, and F1-score compared to the other pre-trained models. Its high performance across multiple evaluation metrics indicates its effectiveness in classifying the dataset.

Table 6 Evaluation performance of individual pre-trained models.

Pre-trained Models	ACC	SEN	PRE	F1-score	
DenseNet201	0.9738	0.9816	0.8865	0.9223	
VGG16	0.9641	0.9553	0.8948	0.9194	
GoogleNet	0.9641	0.9759	0.8956	0.9265	
EfficientNetB7	0.8923	0.8534	0.7687	0.7914	
InceptResNetV2	0.9706	0.9636	0.9244	0.9415	
Xception	0.9608	0.9649	0.8774	0.9093	

(2) Ensemble models classification results

Two ensemble learning setups were formulated and executed based on the performance of the previously evaluated individual deep learning models. These ensembles are identified as Ensemble A (consisting of DenseNet201, GoogleNet, and InceptionResNetV2) and Ensemble B (comprising DenseNet201, GoogleNet, and VGG16). The classification efficacy of each ensemble model is also presented in Table 7. Additionally, the evaluation of classification performance encompasses metrics such as accuracy, sensitivity, precision, and F1-score.

Table 7 Classification evaluation performance for Ensemble A and Ensemble B.

Pre-trained models	ACC	SEN	PRE	F1-score	
Ensemble A	0.9870	0.9718	0.9860	0.9947	
Ensemble B	0.9739	0.9695	0.9718	0.9703	

(3) The proposed hybrid model classification results

According to the findings of the prescribed ensemble learning inquiry, Ensemble A, comprising DenseNet201, GoogleNet, and InceptionResNetV2, demonstrated superior classification efficacy compared to Ensemble B. As delineated in Table 7, the proposed HWST model exhibited notably enhanced classification proficiency, achieving an overall accuracy of 99.34%. This indicates that the model based on Ensemble A, improves collective classification performance by approximately 0.659% in comparison to the individual Ensemble. The identification evaluation performance of HWST model is detailed in the below.

First, Fig. 7 shows the graph of both accuracy and loss during the training of the model. After the initial epochs, the model seems to reach a state of convergence, where both accuracy and loss do not show significant changes. This suggests that the model has reached an optimal or near-optimal state of learning. The high accuracy and low loss indicate a model that is well-fitted to the data, though the early spikes in validation loss suggest that it might have gone through moments of potential overfitting or instability before stabilizing.

Figure 7 The training and testing accuracy and loss for the proposed hybrid Transformer Encoder model.

The classification report of HWST model illustrates its performance in classifying brain tumor categories. Across all metrics (precision, recall, F1-score, and accuracy), the model demonstrates high precision values, indicating minimal false positives. The balanced F1-scores show the model’s ability to minimize both false positives and false negatives. For meningioma, the model achieves a precision, recall, and F1-score of 0.9930, indicating that 99.30% of instances classified as meningioma were indeed meningiomas. This high precision is matched by an equally high recall, meaning the model correctly identified 99.30% of actual meningioma cases. The F1-score, which balances precision and recall, is also 99.30%, affirming the model’s robust performance for this class. There were 143 instances of meningioma in the dataset.

Similarly, for glioma the model demonstrates strong precision, recall, and F1-score values of 0.9859. This indicates that approximately 98.59% of instances classified as gliomas were accurate, with the same percentage of actual glioma cases being correctly identified. The F1-score of 0.9859 further confirms the model’s effectiveness in capturing instances of glioma.

For pituitary tumors, the model achieves perfect precision, recall, and F1-score values of 1.0000. This indicates that all instances classified as pituitary tumors were indeed pituitary tumors, with the model correctly identifying all instances of this class. The perfect F1-score underscores the model’s flawless performance for pituitary tumors.

In addition, the ROC curve analysis for all classes yielded an AUC value of 1, as shown in Fig. 8. This perfect AUC score indicates that the model achieved optimal discrimination between positive and negative instances across all classes. Regarding computational efficiency, the total training time for the model was 7,695.43 s, and the total predicting time on unseen data was 17 s.

Figure 8 Evaluation results of the hybrid Transformer Encoder model in terms of ROC.

Finally, the confusion matrix provided in Fig. 9 illustrates the performance of a classification model in identifying three types of brain tumors. The model shows a high degree of accuracy as indicated by the concentration of higher numbers along the matrix’s diagonal, which represents correct predictions. Specifically, it correctly classified 142 meningiomas, 70 gliomas, and 93 pituitary tumors, with only one case each of meningioma and glioma being misclassified.

Figure 9 Confusion matrix for the model.

To further validate the model’s performance and interpretability, we employed Gradient-weighted Class Activation Mapping (Grad-CAM). Grad-CAM generates visual explanations by highlighting the regions of the input images that were most influential in the model’s decision-making process. This technique provides insight into which areas of the brain tumor images contributed to the classification, helping to ensure that the model’s predictions are based on meaningful features and supporting the reliability of its diagnoses. The Grad-CAM visualizations are presented in Fig. 10, illustrating the regions of interest for each tumor type, which corroborate the model’s accurate performance as indicated by the confusion matrix.

Figure 10 Grad-CAM visualizations of the regions of interest.

The efficacy of our HWST model for brain tumor classification was evaluated against existing methods and demonstrated significantly higher accuracy levels. Table 8 presents the comparative study of brain tumor classification. Our model achieved the highest accuracy among all compared methods when evaluated on the same dataset (Cheng, 2017) with an equivalent number of images (3,064), ensuring a fair comparison of predictive efficiency.

Table 8 Comparison of the proposed model with state-of-the-art methods using the Cheng dataset.

Author	Method	Accuracy	
Polat & Gungen (2021)	ResNet50	99.02%	
Ismael, Mohammed & Hefny (2020)	ResNet50	99%	
Rehman et al. (2020)	Fine-tune VGG16	98.69%	
Swati et al. (2019)	VGG19	94.82%	
	ViT-B/16	97.06%	
	ViT-B/32	96.25%	
Tummala et al. (2022)	ViT-L/16	96.74%	
	ViT-L/32	96.01%	
	Ensemble ViT	98.70%	
Ferdous et al. (2017)	LCDEiT	98.11%	
Proposed model	–	99.34%	

Application and economic benefits for society

This research introduces a methodological approach for early brain tumor detection using MRI image analysis and a self-attention mechanism. The proposed model shows promise in improving the accuracy of tumor identification and classification across various types. It has the potential to significantly impact neuro-oncology by enhancing diagnostic tools and aiding in precise tumor classification. Additionally, this model could advance brain tumor research, assist in image-guided surgeries, and support the development of targeted treatments and therapies.

Conclusion

The study introduces a novel approach to brain tumor classification, combining a hybrid CNN model with Transformer Encoder architecture and ensemble learning techniques. This innovative framework aims to enhance the accuracy and reliability of early detection of brain tumor diseases in MRI images. The hybrid CNN model was evaluated in both binary and multiclass scenarios, demonstrating its versatility and effectiveness across various diagnostic contexts. A comprehensive study using three public MRI datasets confirmed its reliability and generalizability, while the evaluation of various pre-trained models provided insights into optimal architectures. Moreover, Bayesian optimization automated the hyperparameter tuning process, streamlining it and enhancing the model’s efficiency in real-world applications.

In conclusion, the innovative hybrid CNNs model offers a promising avenue for advancing brain tumor classification techniques. Leveraging Transformer Encoder architecture, ensemble learning, and automated hyperparameter tuning, the proposed framework achieved a classification accuracy of 99.34% with a loss of 0.073 using the Cheng Brain Tumor Image dataset from Figshare.

Future works

In the future, there are plans to enhance the proposed deep learning (DL) model with advanced hybrid ensemble methods using new MRI datasets. The proposed approach will integrate explainable artificial intelligence (AI) techniques, such as Local Interpretable Model-agnostic Explanations (LIME) and SHapley Additive Explanations (SHAP), to provide a comprehensive overview of the decision-making processes used by the DL model. This will help increase both patients’ and physicians’ trust and confidence in the diagnostic process. Additionally, incorporating segmentation techniques will further refine the model’s accuracy and reliability. Gathering more data to generalize and improve the model’s performance will be crucial for achieving robust and reliable diagnostic outcomes.

Supplemental Information

Code S1 Code

Additional Information and Declarations

Competing Interests

Author Contributions

Data Availability

The authors declare there are no competing interests.

Nawal Benzorgat conceived and designed the experiments, performed the experiments, analyzed the data, performed the computation work, prepared figures and/or tables, authored or reviewed drafts of the article, and approved the final draft.

Kewen Xia conceived and designed the experiments, performed the experiments, analyzed the data, performed the computation work, prepared figures and/or tables, authored or reviewed drafts of the article, and approved the final draft.

Mustapha Noure Eddine Benzorgat analyzed the data, authored or reviewed drafts of the article, and approved the final draft.

The following information was supplied regarding data availability:

The Brain Tumor Image Dataset is available at Kaggle: https://www.kaggle.com/datasets/denizkavi1/brain-tumor; and figshare: Cheng, Jun (2017). brain tumor dataset. figshare. Dataset. https://doi.org/10.6084/m9.figshare.1512427.v5.

The BT-large-2c dataset is available at Kaggle: https://www.kaggle.com/datasets/ahmedhamada0/brain-tumor-detection.

The BT-large-4c dataset is available at Kaggle: https://www.kaggle.com/datasets/sartajbhuvaji/brain-tumor-classification-mri.

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
