# Peer review of "Enhancing brain tumor MRI classification with an ensemble of deep learning models and transformer integration"

_PeerJ Computer Science, doi:10.7717/peerj-cs.2425_

## Round 0.1 · original submission · Major Revisions

Thank you for submitting your manuscript to PeerJ Computer Science. The review process has been completed, and we have carefully considered the feedback provided by the reviewers.

The reviewers have acknowledged the potential value of your work but have raised several significant concerns, particularly regarding the methodology and experimental evaluation. These concerns require substantial revisions to ensure that the manuscript meets the rigorous standards of our journal.

In light of these comments, I am recommending that your manuscript undergoes a major revision. We encourage you to carefully address each of the reviewers’ comments, paying close attention to the methodological issues and the robustness of your experimental evaluation. A detailed response to the reviewers, explaining the changes made or providing justifications for any unaddressed points, should accompany your revised submission.

Once the revisions have been completed, your manuscript will undergo a further round of review to ensure that all major concerns have been satisfactorily addressed.

We appreciate the effort that you have put into this research and look forward to receiving your revised manuscript.

Reviewer 2 ·

Basic reporting

1. Title is complex - Try to give simple one

2. Literature review is very much odd. I could not find accuracy of some models.

3. Author concentrate only on Brain tumor classification. Why not tumor substrucutres segmentation?

4. Chapter 3 showed like an Imporoved method. Give one name to your method and mention that throughout the article.

5. Type of tumor details are missing in the introduction. Elobrate more about tumor type and how it looks like in the MRI image? Add one more figure like Figure 4 in the introduction section with all type of tumors .

6. How much processing time is reduced after resizing the image?

7. Metrics details are missing in the document.

8. Table 8 alone meaningless

9. Table 9 - Resnet50 accuracy having typo error

10. There is no information on computation time

11. References seems to be old before 2020. Add few more like :
RIBM3DU-Net: Glioma tumour substructures segmentation in magnetic resonance images using residual-inception block with modified 3D U-Net architecture
An Automated Two-Stage Brain Tumour Diagnosis System Using SVM and Geodesic Distance-Based Colour Segmentation
E-Tanh: a novel activation function for image processing neural network models

Experimental design

Hybrid Apporach for classification - Sound

Validity of the findings

no comment

Additional comments

Nil

·

Basic reporting

The manuscript is well written, except for some grammatical mistakes and word repetitions. I urge the authors to thoroughly review the manuscript and correct these issues.

Experimental design

Research question well defined, relevant & meaningful. It is stated how research fills an identified knowledge gap.

Validity of the findings

The proposed work is commendable and innovative, but it lacks validation. Numerous explainable AI models, such as LIME and SHAP, are available. I recommend that the authors validate their model for image classification.

Additional comments

The proposed work is acceptable for publication; however, the methods have been used in other image classification applications. I recommend that the authors validate their model using an explainable AI framework.

---

## Round 0.2 · accepted · Accept

I hope this message finds you well. After carefully reviewing the revisions you have made in response to the reviewers' comments, I am pleased to inform you that your manuscript has been accepted for publication in PeerJ Computer Science.

Your diligent efforts to address the reviewers’ suggestions have significantly improved the quality and clarity of the manuscript. The changes you implemented have successfully resolved the concerns raised, and the content now meets the high standards of the journal.

Thank you for your commitment to enhancing the paper. I look forward to seeing the final published version.

Reviewer 2 ·

Basic reporting

No comment

Experimental design

No comment

Validity of the findings

No comment

Additional comments

All corrections are made.